# Red Arils of *Taxus baccata* L.—A New Source of Valuable Fatty Acids and Nutrients

**DOI:** 10.3390/molecules26030723

**Published:** 2021-01-30

**Authors:** Małgorzata Tabaszewska, Jaroslawa Rutkowska, Łukasz Skoczylas, Jacek Słupski, Agata Antoniewska, Sylwester Smoleń, Marcin Łukasiewicz, Damian Baranowski, Iwona Duda, Jörg Pietsch

**Affiliations:** 1Department of Plant Product Technology and Nutrition Hygiene, Faculty of Food Technology, University of Agriculture in Cracow, Balicka st. 122, 30-149 Cracow, Poland; malgorzata.tabaszewska@urk.edu.pl (M.T.); lukasz.skoczylas@urk.edu.pl (Ł.S.); jacek.slupski@urk.edu.pl (J.S.); 2Institute of Human Nutrition Sciences, Faculty of Human Nutrition, Warsaw University of Life Sciences (WULS-SGGW), Nowoursynowska st.159c, 02-776 Warsaw, Poland; agata_antoniewska@sggw.edu.pl (A.A.); damian_baranowski@sggw.edu.pl (D.B.); 3Department of Plant Biology and Biotechnology, Faculty of Biotechnology and Horticulture, University of Agriculture in Cracow, Al. 29 Listopada 54, 31-425 Cracow, Poland; sylwester.smolen@urk.edu.pl; 4Department of Engineering and Machinery for Food Industry, Faculty of Food Technology, University of Agriculture in Cracow, Balicka st. 122, 30-149 Cracow, Poland; marcin.lukasiewicz@urk.edu.pl; 5Department of Animal Product Technology, Faculty of Food Technology, University of Agriculture in Cracow, Balicka st. 122, 30-149 Cracow, Poland; iwona.duda@urk.edu.pl; 6Institute of Legal Medicine, Medical Faculty Carl Gustav, Dresden Technical University, Fetscherstr. 74, D-01307 Dresden, Germany; joerg.pietsch@tu-dresden.de

**Keywords:** *Taxus baccata* L. red arils, polymethylene-interrupted fatty acids, α-linolenic acid, nutritional value, amino acids, elements

## Abstract

The aim of this study, focused on the nutritional value of wild berries, was to determine the contents of macronutrients, profiles of fatty (FAs) and amino acids (AAs), and the contents of selected elements in red arils (RA) of *Taxus baccata* L., grown in diverse locations in Poland. Protein (1.79–3.80 g/100 g) and carbohydrate (18.43–19.30 g/100 g) contents of RAs were higher than in many cultivated berries. RAs proved to be a source of lipids (1.39–3.55 g/100 g). Ten out of 18 AAs detected in RAs, mostly branched-chain AAs, were essential AAs (EAAs). The EAAs/total AAs ratio approximating were found in animal foods. Lipids of RA contained seven PUFAs, including those from n-3 family (19.20–28.20 g/100 g FA). Polymethylene-interrupted FAs (PMI-FAs), pinolenic 18:3Δ5,9,12; sciadonic 20:3Δ5,11,14, and juniperonic 20:4Δ5,11,14,17, known as unique for seeds of gymnosperms, were found in RAs. RAs may represent a novel dietary source of valuable n-3 PUFAs and the unique PMI-FAs. The established composition of RAs suggests it to become a new source of functional foods, dietary supplements, and valuable ingredients. Because of the tendency to accumulate toxic metals, RAs may be regarded as a valuable indicator of environmental contamination. Thus, the levels of toxic trace elements (Al, Ni, Cd) have to be determined before collecting fruits from natural habitats.

## 1. Introduction

Ample studies have shown fruits to be good sources of phytochemicals and to play a remarkable role in the maintenance of human health as they influence various metabolic processes [1,2]. The composition and quality of fruits derived from natural habitats depend on their genotype, but can also be modified by diverse environmental factors, such as temperature, light, water, soil quality, and altitude [3,4,5]. For example, temperature and light substantially determine the accumulation of soluble carbohydrates in citrus [5]. Cloudberries grown in open habitats differed significantly in their chemical composition from those grown in shaded sites [4]. Apart from cultivated fruits, the wild and underutilized ones also offer some nutritional value, being a rich source of carbohydrates, proteins, fibers, minerals, and vitamins [3,4,6,7,8,9].

European yew (*Taxus baccata* L.) is a non-resinous gymnosperm evergreen conifer tree or shrub up to 15 m in height, common almost all over Europe. It grows naturally at latitudes of up to 63ºN in Norway and Sweden. Large populations of yew grow in Baltic countries, Ukraine, Poland, Romania, Hungary, and Carpathians and Caucasus mountains, and also in Southwest Asia and northwest Africa [10,11]. Unlike many other conifers, the common yew does not actually bear its seeds in a cone. Instead, female yews have red fleshy, berry-like structure around the seeds, known as red arils (RAs), and are open at the tip, which is equivalent to the fruit pulp of many deciduous trees. In Poland, the European yew is under species protection and is listed in the Red Book as a plant at risk of extinction [12].

The genus *Taxus* has generated considerable interest due to its content of diterpene alkaloids known as taxines [13]. Taxine B, which is detected in all parts of yew plants except RAs, is the major compound of the alkaloid fraction (approximately 30%) responsible for their toxicity. Another taxane compound, paclitaxel (taxol A), which is less polar than taxines and has cytotoxic and anticancer activities, is used in cancer therapy (lung, ovarian neoplasia, breast, metastatic carcinoma) and in the second-line treatment of AIDS-related Kaposi’s sarcoma [14,15]. In addition, 10-deactetylbaccatin III, a non-alkaloidal diterpene, contains the fundamental piece of paclitaxel structure (the core taxine ring), thus inducing apoptotic cell death of cancer cells [16].

In different parts of various *Taxus* species, other active compounds such as phenolic constituents (3,5-dimethoxyphenol, myricetin, bilobetin), 50 lignans including neolignas, were identified [15]. These compounds show antibacterial, antifungal, antioxidant, and antiulcerogenic activities [17]. Strong proapoptotic activity of methanolic extract of leaves was confirmed in studies on human cell lines (colon cancer HCT, 116) [11].

It should be pointed out that in the *Taxus* species, only the seedless red fleshy part of berries, RAs, are free of toxic compounds [11,13,15]. Moreover, Siegle and Pietsch [18] revealed in the RAs of yew berries the presence of anticancer and antioxidative taxoid compounds (terpenes and phenolic compounds), with major, however trace share, of 10-deactetylbaccatin III. RAs are an enticing delicacy for many animal species, being a little expressive, slightly sweet, having a bland taste and aroma, and rich in mucous compounds. Besides, they contain a substantial amount of dietary fiber (7.7–10.6 g/100 g) [19]. However, to the best of our knowledge, published data about macro- and micro-nutrient composition of RAs is lacking.

Earlier studies on lipids of seeds of Conifer species, also of those from *Taxacea* family, showed that lipid fraction of seeds was distinguished by a substantial presence of polymethylene-interrupted fatty acids (PMI-FAs), also called Δ5-olefinic acids [20,21,22]. Fatty acids (FAs) from that group frequently bear the first double bond on C5, separated by five methylene units from the next double bond [23]. The chemical structure of PMI-FAs is “uncommon” as compared with polyunsaturated FAs (PUFAs) with a regular position of double bonds, e.g., linoleic and α-linolenic acids in plants [23]. However, they are typical for *Taxus* gymnosperm PMI-FAs [23]. Several pharmacological effects, e.g., modulation of immune response, suppression of hypertension, hyperlipidemia and enhancement of memory acquisition in the central nervous system, were reported for oils derived from conifer seeds containing PMI-FAs [24]. Recent studies on animal models and cell lines showed that the juniperonic acid (20:4Δ5,11,14,17) exerted anti-inflammatory effects [25]. Chen et al. [26] demonstrated that pinolenic acid (18:3Δ5,9,12) could act as a potential anti-cancer agent, reducing the risk of breast cancer by effectively antagonizing prostaglandin E_2_ and cyclooxygenase expression.

Generally, fruits are not a good source of lipid fraction. However, some wild berries revealed to be a good source of lipids and beneficial FAs. For example, *Zantoxylum* fruits and wild sea buckthorn (*Hippophae rhamnoides*), contain substantial amounts of 16:1, *Rhus triparrtium* fruits having 3.8 to 6.4% lipids with substantial presence of PUFA from n-6 and n-3 families, such as *Arbutus unedo* L. berries [27,28,29,30]. Red berries of *Taxus baccata* may thus be regarded as a good source of a range of unsaturated FAs, e.g., PMI-FAs and other.

Thus, as a part of the ongoing interest in the nutritional value of wild fruits, the aim of this study was to determine the contents of macronutrients, profiles of fatty acids and amino acids, and the contents of selected elements in red arils of *Taxus baccata* L., grown in diverse locations in Poland.

## 2. Results and Discussion

### 2.1. Taxus Compounds

Five Taxus compounds were detected in RAs samples. Their contents significantly varied between collection sites (Table 1). Two of those compounds dominated: 10-deacetylbaccatin III and baccatin III, with the highest shares being found in samples from Zielona Gora sites, as compared with other three sites. This could have been due to the higher annual temperature and exposure to UV radiation at Zielona Góra site as compared with other locations (Table 2, Figure 1). The effects of light intensity and temperature on taxane concentrations in needles and twigs was reported [31]. Previous studies also revealed seasonal differentiation in taxane content, e.g., 10-deacetylbaccatin III, baccatin III and cephalomannine in seeds [14,18,32]. These non-alkaloid diterpenoid compounds are appreciated because of their proved cytotoxicity to cancer cells [13,16,33]. These results were comparable with those reported for RAs by Siegle nad Pietsch [18]. Much higher contents of 10-deacetylbaccatin III, baccatin III were detected in leaves of *Taxus baccata* L. and of other *Taxus* species (6 to 10-fold more) and in twigs of other *Taxus* species (25-fold more) [14,34].

Other three compounds, especially taxol A (paclitaxel) were detected only in trace amounts in RAs samples (0.02–1 µ/g) in contrast with other parts of *Taxus baccata* plant, e.g., leaves, which contained 12.8 to 33.7 µ/g of paclitaxel, depending on the month of collection [14]. In leaves and twigs of other *Taxus* species, the range of amounts of paclitaxel amounted to 23.8–150 µ/g. It is worth mentioning that in the studied RAs samples, no traces of taxine alkaloids responsible for toxicity (taxines A and B) were detected. We thus consider recommending analyzing the nutritional composition of RAs as a fruit of potential value in human nutrition.

### 2.2. Proximate Composition

The results (Table 3) revealed a significant (*p* < 0.05) variation in the nutrient contents of RAs collected from different sites. As pointed out by Hegazy et al. [9], the moisture content affects many physical properties of fruits, such as viscosity, weight, and density, and is a helpful indicator during fruit harvesting, storage, and processing. RA samples had a higher moisture content (75.8%, on average) than wild berries from the Mediterranean region: strawberry-tree berries, blackthorn, and rose (48.7–60.9%), but lower than cultivated fruits: cherries and red raspberries (86.4–92.7%) [6,35]. As shown in Table 3, significant differences (*p* < 0.05) were noted in the fruit moisture from different locations, most likely due to different environmental conditions, such as water availability, sunlight, and wind exposition [36].

The protein content of RA (1.79–3.80%) was higher than in cultivated fruits (0.48–1%; cherry, blueberry, strawberry, red raspberry) [35]. In addition, RAs exceeded some species of wild fruits, e.g., mulberries, in protein content, but was about twice lower than in *Rhus tripartitum* fruits derived from two different locations in Tunisia [2,27].

Site differentiation in protein content of RAs was confirmed in previous papers on berry fruits and strawberry-tree fruits; it was suggested that protein content of fruits can vary with soil and climatic conditions [27,36].

Carbohydrates were the main macronutrient in RAs, accounting for 18.43 to 19.30 g/100 g, i.e., much more than in cultivated berries, such as strawberries, red raspberries, and blueberries (6.30–11.54 g/100 g), and in mulberries [2,35]. In turn, a higher content of carbohydrates than in RAs was assayed in wild berries [6]. The chromatographic analysis revealed the predominant share of fructose (5.36–6.43 g/100 g) followed by glucose and sucrose, irrespective of collection site (Table 4). Similar quantitative shares of sugars were confirmed in different wild berries (strawberry-tree, blackthorn, rose fruits) [6]. The sucrose content of RAs was significantly (*p* < 0.05) differentiated between all locations. The lowest content of sucrose (0.92 g/100 g), together with highest amount of fiber (10.6 g/100 g), was found in RAs collected at the Koszalin site [19]. Samples from the other three sites contained higher amounts of sucrose (1.65–2.65 g/100 g), most likely because of differences in temperature during ripening (Table 2). Our results contrasted with the data of Zheng et al. [5] about the contents of soluble sugars in *Lycium barbarum* berries; they found that genetic factors and the degree of maturity had a larger effect on sugar contents than environmental factors.

Generally, RAs contained a substantial amount of lipids, reaching 1.39 to 3.55 g/100 g (except those from Warsaw site). Our results opposed the view that fruits were generally a poor lipid source, such as *Rosa rugosa* pericarp (0.67–0.88%), mulberry species (0.14–0.40%), blackberries, red raspberries, and strawberries (0.25–0.42%) [2,35,37]. However, berries of some species: Goji berries, wild fruits from Saudi Arabia, are rich in lipids (2.23–5.5%) [9,38]. Total carbohydrates, simple sugars, and proteins are the vital nutrients in many fruits, as they are the main source of energy [9]. Our results also proved RAs to be an important source of lipids.

The marked differentiation in lipid contents in RAs samples derived from different sites is in agreement with previously reported data concerning other berries, namely wild *Arbutus unedo* (0.72–1.66%), mulberries (0.14–0.40%) [2,30]. That discrepancy may be due to different environmental conditions [30].

On the basis of the proximate analysis, a portion of 100 g of RAs provides, on average, 106 kcal. It is about 3.5-fold lower than the energy value of other wild berries (strawberry-tree, blackthorn, rose), most likely because of its low sugar content [6]. Hence, RAs can be recommended as a low-calorie snack.

The ash content of RAs, reaching 0.44 g/100 g (Table 3), was about twice higher than that in cultivated fruits (blackberries, red raspberries, and strawberries) and similar to that of *Prunus avium* L. [35,39]. The contents of both dry matter and ash in plants are usually affected by the climate and soil conditions [3].

### 2.3. Amino Acid Profile

Eighteen AAs were detected in RAs samples. Ten of them were essential AAs (EAAs), and seven non-essential AAs (non-EAAs). The contents of AAs varied significantly (*p* < 0.05) depending on the growth site (Table 4). For example, the differences in total EAAs were remarkable, ranging from 720 mg/100 g to 1207 mg/100 g (Cracow and Warsaw sites, respectively). A similar range of total EAAs (TAAs) was found in berries of Rosa *roxburghii* and Rosa *sterilis* and *Rhoodomyrtus tomentosa* (sim fruits) [8,40]. Environmental variation in AAs was also found in goji berries in China. The authors stated that alkaline soil and large day/night temperature difference were optimal for wolfberry fruit production [7].

The EAAs/TAAs ratio in RA amounted to 43%, on average. According to FAO and WHO, foods with EAAs/TAAs ratio above 40% are an ideal protein source. Generally, foods of animal origin, such as eggs, milk, and fish, have EAAs/TAAs ratios of 41 to 46% and are sources of high-quality protein [2]. The EAAs/TAAs ratio of RA approximates that found in animal foods, suggesting that the RAs could be also used as a source of high-quality protein in human diet.

Leucine was the major component of EAAs, followed by lysine and valine. Predominating shares of lysine and leucine were also found in wild fruits collected from Saudi Arabia [9]. The contents of leucine in RA (123–220 mg/100 g) were 2.5 to 3-fold higher than in Rosa *roxburghii* and Rosa *sterilis* berries [8]. The content of lysine, which is especially important for growing organisms, was similar to that found in rose hips [8]. The content of branched-chain AAs (BCAAs: leucine, isoleucine, and valine) in food is especially important, as these are closely related to human health. Apart from being the building blocks of proteins, they control the protein and energy metabolism, and serve as amino-group donors to synthesize glutamate in brain [41]. The share of BCCAs in total AAs of RAs was impressive, reaching 18.4% on average, and was similar to that found in animal proteins (about 20%) [41]. Hence, we suppose that RA can serve as a BCCAs-rich new ingredient in diet.

Glutamic acid was the major component of the non-EAAs, with 1.5 to 2.5-fold higher content in RAs from Warsaw site (510 mg/100 g) compared with samples from the other three sites (200–317 mg/100 g). The prevailing share of glutamic acid in AAs of wild fruits was also confirmed by Guo et al. [7]. The major AAs from the non-EAAs group in RA included serine (153–237 mg/100 g) and proline (150–220 g/100 g).

### 2.4. Fatty Acid Composition

The application of high-resolution GC enabled detecting 25 fatty acids (FAs) in lipid fraction extracted from RAs. The contents of many of them differed significantly (*p* < 0.05) depending on the fruit collection site (Table 5). Saturated FAs (SFAs) were represented mainly by palmitic (20.43–24.37 g/100 g FA) and myristic (6.76–10.76 g/100 g FA) acids. The total content of other six SFAs was low and did not exceed 3 g/100 g FA. A similar content of C16:0 was found in *Rosa rugosa,* but a much higher one in sea buckthorn pericarps [42]. The presence of palmitic and myristic acids in *Taxaceae* species (seeds) was confirmed in a previous study; however, their contents were much lower than in RA from our study [22].

The total content of MUFAs RAs lipids varied from 7.69 to 13.04 g/100 g FA. The same trend was reported in the content of the major MUFA—oleic acid. In contrast to the lipid composition of other berry fruits, MUFAs were the least significant FAs in the RAs [38,42,43]. Lipid fraction of RAs from Koszalin site had about twice higher content of oleic acid compared, with samples from Cracow and Warsaw sites (Table 2). It is in accordance with the study of Issaoui et al. [44], who found that Tunisian olive oils from the north locations showed greater content of oleic acid comparing with samples from the south.

The lipid fraction of RAs was extremely rich in PUFAs (10 compounds). Among them, five compounds belonged to the n-6 family, and two to the n-3 family. The major PUFA was linoleic acid (30.92 g/100 g FA); it was the most abundant in the lipid fraction of Zielona Góra-site samples. The α-linolenic was the second important PUFA, especially in samples from the other three locations (23.43–26.50 g/100 g FA in average; Table 5). It should be noted that similar to *Rosa rugosa* fruits, the RAs may be perceived as a valuable source of α-linolenic acid belonging to the n-3 family [37]. Lipids of RAs also contained a long-chain PUFA (LC-PUFA) trienoic acid (20:3Δ11,14,17) from the n-3 family (Table 5), the necessary precursor of juniperonic acid [23]. That compound is rather unusual for vegetable oils.

Among other PUFAs, three PMI-FAs (18:3Δ5,9,12; C20:3Δ5,11,14 and 20:4Δ5,11,14,17), unique for the gymnosperm plants, were identified [20,21,22,23]. Even though pinolenic acid does not belong to essential FAs, it forms biologically active metabolites in the presence of cyclooxygenase or lipoxygenase, and these metabolites can partially relieve some of the symptoms of essential FAs deficiency [45]. Focusing on PMI-FAs, sciadonic dominated quantitatively, its content in RAs lipids from Zielona Góra site being higher compared with other sites. The highest content of pinolenic acid was found in lipids of RA derived from Cracow site (0.23 g/100g FA). Those results are in accordance with previously reported in lipids of *Taxus baccata* seeds [20,22]. However, our study did not confirm the presence of taxoleic acid (18:2Δ5,9), a typical FA for lipids of *Taxus baccata* seeds (Appendix A) [20,21,22]. Substantial differences between the FAs profile of pericarp and seeds of fruits confirmed a previous study on *Garcinia* fruits [46]

There are several factors that can affect FAs composition including plant origin, environmental conditions and temperature, throughout the time between flowering and ripening [47]. For example, low temperature promotes the synthesis of PUFAs in plants, especially the LC-PUFA ones [29]. This was noted in lipids of RAs from Koszalin site, having the highest share of four LC-PUFA among studied samples (Table 5).

Contrary to expectations, lipids of RAs samples derived from Zielona Góra site, which had better climatic conditions (temperature) than other locations, contained a significantly higher amount of total PUFA (54.4 g/100 g FA), as compared with samples of the other three sites (50.7 g/100 g FA on average; Table 5), most likely due to better parameters of the brown soil there (Table 2). An impressive amount of α-linolenic acid in RAs from the Cracow sample could be attributed to favorable soil conditions and high precipitation, appropriate for the demanding *Taxus baccata* trees [48].

For nutritional reasons, it is essential to search for sustainable vegetable sources of PUFAs, especially for α-linolenic acid from the n-3 family. The primary biological role of α-linolenic may consist of it being a substrate for long chain PUFAs in EPA and DHA synthesis [49,50]. Baker et al. [49] pointed out in their review that epidemiological studies in Europe, USA, and Japan indicated a decreased risk of CVD and inflammation with increasing consumption of long-chained n-3 PUFAs. The lack of α-linolenic provision in the diet decreases the availability of DHA for incorporation into neural and retinal membranes and may explain the impact of α-linolenic deficiency on vision [50]. From the consumer’s point of view and for nutritional reasons, the contents of PUFA were computed per 100 g of RAs (Table 6). The differences in PUFAs contents were related mainly to lipid content (Table 3), thus the portion of 100 g RAs from Koszalin and Cracow sites differed by a 2.5 to 4-fold higher content of PUFA compared with samples from the other two sites. In addition, RA from Koszalin and Cracow sites had high amounts of unique PMIs, especially sciadonic (on average, 25.5 mg/100 g RAs; Table 6).

Except Warsaw location, RAs proved to be a substantial source of linoleic (n-6 PUFA; 429–757 mg/100 g RA). High blood levels of n-6 acids were considered an increased risk of inflammatory and allergic conditions in epidemiological studies [48]. An increased intake of α-linolenic from the diet has the potential to limit the production of n-6 derived proinflammatory mediators and to enhance the biological efficacy of long chain n-3 PUFA [50]. As shown in Table 6, samples of RAs from two sites were an especially valuable source α-linolenic FA; 100 g of fruits may provide 832 to 938 mg of 18:3Δ9,12,15 acid. It should also be pointed out that the beneficial ratio of n-6/n-3 PUFAs in RAs is 1:0.8–1:1.7, as shown in Table 6. This ratio was much lower compared with the current ratio n-6/n-3 PUFAs of Western diet, from 15:1 to 16.7:1 [49].

Based on these results, RAs may represent a novel vegetable dietary source of valuable PUFAs belonging to n-3 family, including the long-chain ones and also unique PMI-FAs.

### 2.5. Elemental Characteristic

The contents of macroelements (K, P, S, Ca, Mg, Na), microelements (Zn, Fe, B, Cu, Mn, Cr, Mo, Co), and metals (Al, Ni, Bi, Ba, In, Ti, Li, Ag, Cd, Ga), are presented in Table 7. Most of them were significantly (*p* < 0.05) dependent on the site of RA collection.

Potassium (K) was the most abundant element in RA (772–878 mg/100 g), followed by P, S, Ca, and Mg. It was about 2 to 3-fold higher than reported for five species of wild fruits in Saudia Arabia and mulberries in China [2,9]. A much higher amount of potassium than in RAs was found in goji berries (2100 mg/100 g) [43]. RAs had potassium content comparable to that of bananas, regarded as a typical potassium source in the diet [51]. Potassium is an essential mineral, important to maintain body water and to participate in transmitting nerve impulses to muscles [51]. An adult human being needs approximately 4700 mg K per day, thus, RAs consumption can meet the daily required amount [52].

RAs contained, on average, about 100 mg/100 g phosphorus (P), which does not meet the recommended daily allowance (RDA) for adults [52]. In addition, the content of sodium (Na) was low (0.86 to 4.90 mg/100 g). However, it should be emphasized that P and Na consumption in developed countries exceeds RDA mainly due to the nearly ubiquitous distribution of phosphorus-based food additives [53].

The content of calcium (Ca) in RAs (about 20 mg/100g) was much higher than in blackberries, red raspberries, strawberries, and cherries [35]. However, mulberries and goji berries have more reach in calcium than RAs (71–124 g/100 g and 126–149 mg/100 g, respectively) [2,43].

Despite the physiologic role of magnesium (Mg) and its proven or potential benefits, its dietary intake is known to be inadequate in many populations [54]. The abundance of magnesium in RAs (23 mg Mg/100 g, on average) allows meeting only 7% of its RDA for a healthy adult, as in the case of mulberries [2], while the consumption of goji berries contributed to 15% of RDA of magnesium [43].

With regard to microelements, irrespectively of the collection site, RAs had a substantial amount of zinc (Zn; 948–1507 µg/100g), similar to the wild bilberries from the Eastern Italian Alps, but much more than many cultivated berries [3,34,43]. Zinc is necessary for many enzymatic reactions and for the absorption of B-group vitamins. It allows maintaining healthy skin, self-immunity, and good functioning of the prostate gland [54]. Consumption of a 100 g portion of RAs allows meeting from 11 to 15% of the RDA of Zn for adults, depending on sex.

Samples of RAs differed significantly (*p* < 0.05) in iron (Fe) content (976–2537 µ/100 g) depending on the collection site (Table 2). Iron, as the constituent of hemoglobin, myoglobin, and of many enzymes, is an essential nutrient. Its adequate supply is especially important for females aged 14 to 50 years. Consumption of a 100 g portion of RAs allows meeting from 9 to 15% of the RDA for adults. However, according to literature, 100 g serving of other berries (blueberry, blackberry and goji berry) cover a higher contribution of RDA of iron (21 to 90%) than in the case of RAs [35,43].

Manganese (Mn) is required for macronutrient metabolism. No formal RDA for Mn was established, but the US NRC set an estimated safe and adequate dietary intake of 2 two 5 mg/day for adults [55]. RAs differed (*p* < 0.05) in Mn content between collection sites: much higher levels were noted in samples from Koszalin and Cracow sites (521–722 µ/100 g) than from Warsaw and Zielona Góra sites (76.5–103.9 µ/100 g). Generally, as compared with literature data about Mn content in other fruits (wild strawberry-tree fruits), RAs may be regarded as a good source of Mn [36]. However, as compared with RA, much higher Mn content was found in wild bilberries, most likely because of the specific composition of soil in the natural habitat of the Eastern Italian Alps [3]. In addition, goji berries, organically grown using organic fertilizers, were rich in Mn (980 g/100 g) [43].

The content of copper in RAs (225 µg/100 g, on average) was about twice higher than in mulberries, but much lower than in goji berries and *Rosa sterilis* fruits [2,8,43].

Boron (B) was the third quantitatively important element in RAs (464–1152 µg/100 g) and proved to be a rich source of boron, similar to many popular nuts (Brazil, pistachio, cashew) [56]. Boron plays an important role in osteogenesis, its deficiency was shown to adversely impact bone development and regeneration, and to support the effects of estrogen, testosterone, and vitamin D. It was suggested that humans need at least 0.2 mg/d of boron, and that the diet should provide approximately 1 to 2 mg B/d [56].

The presented results are thought to reflect soil types and properties of natural sampling sites, as some elements (Fe, Mn, Cu) are abundant in podsols and brown acid soils [3,8]. Micronutrients may vary largely depending on environmental conditions, such as precipitation, humidity and soil composition, as they could induce responses to physiological stress, when the minerals could act as cofactors regulating the metabolic pathways of the plant [36].

The contents of chromium (Cr), molybdenum (Mo), and cobalt (Co) were relatively low (Table 7). The minimum RDA for Cr amounts to 24 µg/day for most adults [57]. Consumption of a 100 g portion of RAs meets 30 to 50% of its RDA. Shim fruits (*R. tomentosa*) grown in Vietnam is a better source of Cr than RAs [40]. Cobalt is included in vitamin B_12_-cobalamine and plays a very important role in the synthesis of AAs and some proteins in nerve cells, and in producing neurotransmitters. The RDA of Co is 5 µg/day. The content of Co in a 100 g portion of RAs exceeded the RDA, except for the samples from Koszalin site (Table 7).

Regarding the presence of metals, the content of aluminum (Al) dominated among the detected metals in RAs (Table 7). Anthropogenic sources of many metals in soils are natural processes (e.g., weathering of rocks), mining and smelting activities, use of sewage sludge and phosphate fertilizers, which may contain heavy metals as impurities. It should be pointed out that the accumulation of trace metals is a normal and essential process for the growth and nurturing of plants [57]. Thus, RAs grown at the Koszalin site on brown acid soil contained 3- to 6-fold more aluminum (Al) than RAs from other sites (Table 7) [3]. The Al content in e.g., bilberries, lingonberries, and rosehips in Finland was higher than in RAs [58]. Since Al is toxic, the EFSA established a Tolerable Weekly Intake (TWI) of 1 mg Al per kg of body weight [59].

RAs from the Koszalin site contained a 2- to 4-fold higher amount of nickel (Ni) compared with samples from other sites, most likely due to environment contamination [57]. Contents of Ni in RAs collected from other sites were similar to that found in many fruits [40,57,58].

Cadmium (Cd) is a heavy metal, especially toxic for kidneys, but may also induce bone demineralization, and was classified as carcinogenic to humans. Cereals and cereal products, vegetables, nuts and pulses, starchy roots and potatoes as well as meat and meat products, contribute most to human exposure. The EFSA has set TWI for cadmium at 2.5 µg Cd per kg of body weight (µg/kg BW) [60]. Given the health effects of cadmium on humans, its maximum level in fruits and vegetables was set at 0.05 mg/kg, accordingly EC 1881/2006 [61]. Depending on the site of fruit collection, RAs contained 4.78 to 24.66 µg Cd/100 g. The samples from Cracow site exceeded the acceptable Cd level almost five times (Table 7), whereas in the samples from Koszalin site, its content was below the acceptable level, similar to freeze-dried strawberries in China [1]. In addition, it is noteworthy that the contents of other trace metals (Bi, Ba, In, Ti, Li, Ag, Ga) were below 80 μg/100 g, i.e., under their maximum permissible limits [3,62]. No Pb was detected in the RAs samples.

The presence of trace metals in fruits may be attributed not only to the natural background of heavy metal content in the soil geochemistry but also derived from the environmental pollution [40,57]. Thus, the growing environment and in particular the soil aluminum, cadmium and nickel concentration should therefore be taken into account when choosing harvest region [40].

## 3. Material and Methods

### 3.1. Sample Collection

Red berries of *Taxus baccata* L. were collected from plants growing in natural habitats in four different sites of Poland (West, Central, North and South), in the neighborhood of cities: Zielona Góra, Warsaw, Koszalin, and Cracow, respectively (Figure 2). In each site, red berries were harvested thrice (throughout September to October, 2018), from 10 trees each (from different parts of crown), growing in three places (*n* = 9). The plants were identified as *Taxus baccata* L. by morphologic comparisons of leaves, flowers, buds, bark of trees and berries, according to Seneta [63] and Krüssmann [64]. Soil and climate conditions at fruit collection sites are presented in Table 2 and Figure 1. Fruits were manually separated from the seeds to obtain RAs for analyses.

### 3.2. Analysis of Taxus Compounds

Sample preparation: aril samples (250 mg each) were dried for 24 h at 60 °C, mixed with 600 µL ammonium buffer (pH = 9), atropine-D3 (Sigma-Aldrich, Steinheim, Germany) as the internal standard, and 1.2 mL of dichloromethane, followed by vortexing for 2 min and centrifuged for 5 min. Next, the organic phase was separated and evaporated to dryness under a stream of nitrogen. The residues were then redissolved in mobile phase (water/acetonitrile, 90:10 *v*/*v*), and 20 μL of the eluate was injected into a HPLC column.

The separation was performed using a Luna Pentafluorophenyl (2) 100 A column (150 × 2 mm, 5 μm; Phenomenex, Aschaffenburg, Germany). LC-MS/MS analysis was conducted on an Agilent 1260 Infinity HPLC system (Agilent Technologies, Santa Clara, USA) coupled to a 3200 QTrap (AB Sciex, Darmstadt, Germany) equipped with an electrospray ionization (ESI) source. The separation was carried out with water/ammonium format (2 mM)/formic acid (0.2%) mixture (solvent A) and an acetonitrile/ammonium format (2 mM)/formic acid (0.2%) mixture (solvent B). The initial solvent ratio was 90:10 (A:B) and was gradually decreased to 10:90 (A:B) within 10 min, the flow rate being 0.5 mL/min. This was held for 5 min and a flowrate of 1 mL/min was applied. Then, the gradient went from 10:90 (A:B) at 15 min back to 90:10 (A:B) at 15.5 min using a flow rate of 1.5 mL/min; the temperature was 20 °C.

The MS source temperature was set to 630 °C, curtain gas to 35 psi, ion source gas 1 to 45, ion source gas 2 to 90 psi, collision gas (CAD) to medium and ion spray voltage to 5500 V. Individual compounds were detected in ESI+ mode and identified by multiple-reaction monitoring (MRM) mode following two mass transitions per analyte [18].

To mimic the red arils matrix, a calibration curve with redcurrant (*Ribes rubrum*) berries was used for quantification; 250 mg of redcurrant berries (dried for 24 h at 60 °C) were spiked with a mix of the five standards (Taxol A, 10-DAB III, BAC III, Cephalomannine and Taxinine M; Sigma Aldrich, Steinheim, Germany) in concentrations ranging from 0.002 to 40 µg taxanes per g, extracted and analyzed as described above.

### 3.3. Analysis of Proximate Composition

Dry matter, ash, and protein contents were determined according to AOAC procedures [65]. Ash content was determined by sample incineration in a muffle furnace (Nabertherm, Germany) at 550 °C. The extraction and determination of lipids from RAs were carried out using the Folch’s method with chloroform-methanol mixture (2:1, *v*/*v*). The total energy content was computed as follows:Energy (kcal) = 4 × (g protein + g carbohydrate) + 9 × (g lipid).(1)

### 3.4. Analysis of Sugars

The RAs were homogenized for one minute (13,500 rpm) in 80% aqueous ethanol, using a DI 25 homogenizer (Ika Warke, Dusseldorf, Germany), and then centrifuged at 2490× *g* for 20 min in an MPW-260R device (Warsaw, Poland).

An HPLC analysis of sugars (glucose, fructose, sucrose) was performed using a Dionex Ultimate 3000 instrument (Thermo Scientific, Germany) equipped with a refractive index detector (RefractoMax 521). Separation of sugars was conducted on a LiChrospher 100-10 NH_2_ (5 µ) column (250 × 4 mm). The isocratic elution mobile phase was provided using acetonitrile/water (87:13 *v*/*v*) at a flow rate of 1.3 mL/min. The identification of sugars was made by comparing the relative retention times of sample peaks with standards (Sigma Aldrich, Poland). Quantification of sugars was made using four-point calibration curves (in a concentration range of 0.1 to 1 mg/mL, for each compound). The contents of sugars were expressed as g/100g of fresh weight of RAs.

### 3.5. Analysis of Amino Acid Profile

Amino acids were quantified by HPLC after an acidic hydrolysis according to Dhillon, Kumar, and Gujar [66] and using an AccQ-Tag reagent kit from Waters (Milford, MA, USA) for derivatization of amino acids.

Each RAs sample (ca. 30 mg) was hydrolyzed with 4 mL of HCl and 15 μL of phenol at 110 °C for 24 h, and then entrapped in N_2_ atmosphere. The hydrolyzate was filtered through syringe filters (0.45 μm), and then dried with N_2_. Next, 10 μL of the sample was mixed with 70 μL of borate buffer (pH 8.2–9.0), then 20 μL of 6-aminoquinolyl-*N*-hydroxysuccinimidylcarbamate acetonitrile solution (3 mg/mL) were added to the mixture (AccQ-Tag reagent kit, Waters, Milford, USA). Analogous procedures were used in the case of standards.

The amino acid profile (AAs) was identified on a Dionex Ultimate 3000 HPLC instrument (Thermo Scientific, Germering, Germany). Separation was provided on a reverse-phase Nova-Pak C18 column (4 μm, 150 × 3.9 mm) (Waters, Milford, MA, USA) at 37 °C. Elution was run in a two-component gradient at 1 mL/min; eluent A: acetic-phosphate buffer, eluent B: acetonitrile-water (60:40). The detector (VWD-3400RS) was set at 240 nm wavelength. The AAs peaks were computed from AAs standards (Sigma-Aldrich, Poznan, Poland) run at five concentrations. Individual AA values were expressed as mg/100 g of fresh weight of RAs.

### 3.6. Analysis of Fatty Acid Composition

Lipid fraction extracted from each RAs sample was used for derivatization of triacyloglycerols into methyl esters of fatty acids (FAMEs) for gas chromatography analysis (GC). Lipids were saponified by boiling in 0.5 mol/L NaOH solution for 10 min. The FAMEs were prepared by transmethylation using a catalyst (95% H_2_SO_4_). Briefly, the samples were heated in a water bath at 100 °C for 40 min in a mixture of sulfuric acid and methanol, followed by the addition of n-hexane. After cooling, saturated NaCl solution was added and mixed thoroughly. Finally, 1 μL of the upper phase containing FAMEs was injected into the chromatograph (GC) injection port.

The FAMEs were analyzed by GC using an Agilent 6890N (HP Agilent, Santa Clara, CA, USA) instrument equipped with a flame ionization detector, a capillary column with the stationary phase of high polarity (100 m, 0.25 mm I.D., film thickness 0.1 µm; Rtx 2330 Restek). The analyses involved a programmed run with temperature ramps. The initial oven temperature was 120 ºC for 40 min, and was then ramped to 155 °C at 1.5 °C/min and held for 50 min. The temperature was then ramped again at 2 °C/min to 210 °C and held for 35 min. Injector and detector temperatures were maintained at 250 ºC; hydrogen was used as the carrier gas at the flow rate of 0.9 mL/min. The peaks were identified by comparison with Supelco 37 No. 47885-U standards and PUFA standards (Sigma Aldrich, Poznan, Poland). Identification of peaks of polymethylene-interrupted FAs was achieved by using chromatograms FAMEs of lipid extracted from *Taxus baccata* seeds (presented in Appendix A) and accordingly published chromatogram [20]. The contents of individual FAs were expressed in g/100 g FAs.

### 3.7. Elemental Analysis

The lyophilized samples of RAs were ground (Fritsch Pulverisette 14, Germany) and microwave-mineralized in a CEM MARS-5 Xpress mineralizer (CEM World Headquarters, Matthews, NC, USA) in HNO_3_ (65%). The contents of macroelements, microelements, and trace metals were determined by the inductively coupled plasma optical emission spectroscopy (ICP-OES) according to Pasławski and Migaszewski [67], using a high-dispersion ICP-OES (Prodigy Teledyne, Leeman Labs, New Hampshire, MA, USA).

### 3.8. Statistical Analysis

All analyses were performed in triplicate. The results were expressed as means and standard errors (SE) and subjected to one-way ANOVA followed by Tukey’s test. Differences between mean values were considered significant at *p* < 0.05. Analyses were performed with Statistica 3.1 software (Statsoft, Inc.,Tulsa, OK, USA).

## 4. Conclusions

Red arils of *Taxus baccata* L. could be used as a vegetable source of high-quality protein with predominating shares of lysine and leucine and can serve as a branched-chain amino acid-rich new ingredient of human diet. Because of the low content of simple sugars, red arils can also be recommend as a low-calorie snack. In vegetarian diet, red arils may be regarded as a source of iron and zinc, providing 9 to 15% of the recommended daily allowance. Red arils of *Taxus baccata* may represent a novel dietary source of valuable PUFAs belonging to the n-3 family, and the unique polymethylene-interrupted FAs, such as pinolenic, sciadonic and juniperonic. In addition, the beneficial ratio of PUFAs n-6/n-3 (from 1:0.8 to 1:1.7), much lower than that in the Western die, t is to be noted. Depending on the location, the consumption of 100 g of red arils would provide 204 to 998 mg PUFAs of the n-3 family. It may thus be worth applying for the GRAS (Generally Recognized as Safe) status of red arils as a safe food additive.

Site differentiation in the contents of macronutrients, fatty acids, amino acids, and macro- and micro-elements in red arils resulted from different environmental conditions such as water availability (sum of precipitation), sunlight intensity, and soil parameters and composition.

Because of the tendency to accumulate toxic metals (Al, Ni, and Cd), red arils may be regarded as a valuable indicator of environmental contamination/pollution. Thus, the levels of toxic trace elements (Al, Ni, Cd) have to be determined before collecting fruits from natural habitats. Although the samples of red arils were free from taxine alkaloids, we recommend monitoring taxus compounds to ensure safety of consumers. 

Full understanding of the nutraceutical potential of red arils requires a further systematic analysis of other fractions, e.g., of phenolic compounds and carotenoids, which is expected to be reported soon.

## Figures and Tables

**Figure 1 molecules-26-00723-f001:**
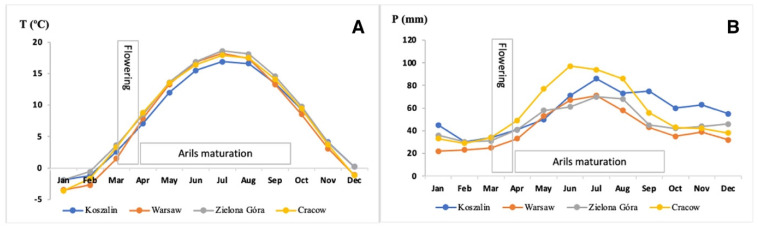
Annual changes in selected climate conditions in diverse locations in Poland: (**A**)—average temperature (°C), (**B**)—total precipitation (mm).

**Figure 2 molecules-26-00723-f002:**
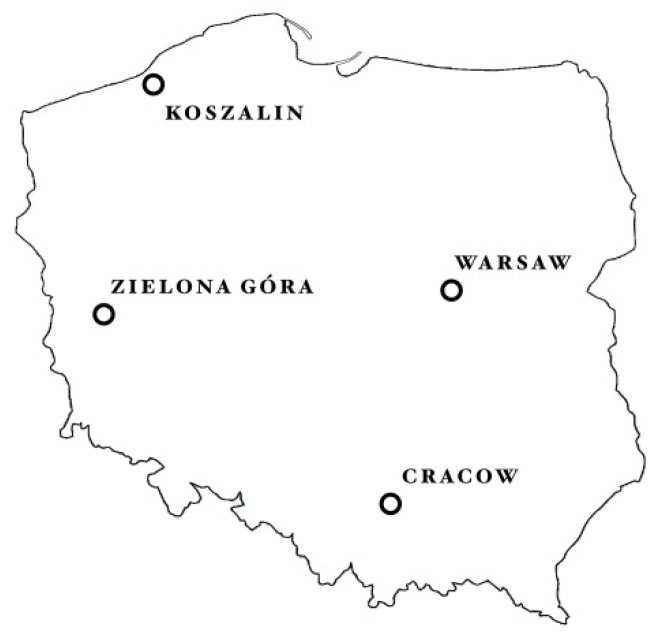
Location of collection of Taxus baccata red berries in natural habitats in Poland.

**Table 1 molecules-26-00723-t001:** Taxus compounds of red arils (µg/g of dry weight, *n* = 9).

Compound	Fruit Collection Site			
	Zielona Góra	Warsaw	Koszalin	Cracow
10-Deacetylbaccatin III	19.80 ± 0.64 ^c^	3.90 ± 0.13 ^a^	7.40 ± 0.29 ^b^	4.10 ± 0.18 ^a^
Baccatin III	6.30 ± 0.13 ^c^	2.00 ± 0.07 ^a^	2.30 ± 0.03 ^a,b^	2.40 ± 0.06 ^b^
Cephalomannine	0.05 ± 0.00 ^b^	0.18 ± 0.01 ^c^	0.12 ± 0.01 ^a^	0.12 ± 0.00 ^a^
Taxinine M	0.13 ± 0.01 ^d^	0.05 ± 0.00 ^c^	0.03 ± 0.00 ^b^	0.02 ± 0.00 ^a^
Taxol A	0.02 ± 0.00 ^b^	0.10 ± 0.00 ^a^	0.05 ± 0.00 ^c^	0.05 ± 0.00 ^c^

^a, b c, d^—values baring the same superscripts in rows do not differ significantly (*p* < 0.05) from each other.

**Table 2 molecules-26-00723-t002:** Characteristics of growth locations of red arils.

Growth Site Characteristics				
Exposition	West	Central	North	South
Location of natural habitats	Zielona Góra	Warsaw	Koszalin	Cracow
Latitude/longitude	51°56′N 15°30′E	52°13′N 21°01′E	54°11′N 16°11′E	50°05′N 19°58′E
Altitude above sea level	140 m	90 m	40 m	240 m
**Soil Parameters**				
Type of soil	brown earths soil	podsolic soil	acidic brown soil	alluvial soil
Soil pH	>5.6	5.26–5.5	<4.5	5–5.25
Mineral nitrogen (mg/kg)	75.1–113.2	<10	20.1–35	20.1–35
Phosphorus availability (mg P_2_O_5_/100 g)	15–20	10–15	>25	10–15
Potassium availability (mg K_2_O/100 g)	15–20	<15	<15	15–20
Magnesium availability (mg Mg/100 g)	6–7	<5	5–6	>10
**Weather Conditions**				
Average annual temperature (°C)	8.8	7.7	7.9	8.2
Average temperature (°C) from flowering to sampling *	13	12.1	11.7	12.6
Total annual precipitation (mm)	572	501	683	678
Average precipitation (mm) from flowering to sampling *	416	385	490	536
Cloudiness (okta)	5.6	5.7	5.9	5.3
Exposure to solar radiation (h)	1700–1800	1700–1800	1700–1800	1600–1700
Intensity of light (W/m^2^)	120–140	120–140	110–130	110–130

°^,^*—from flowering (March–April) until the end of red arils maturation (September–October).

**Table 3 molecules-26-00723-t003:** Proximate composition of red arils (means ± SE, *n* = 9).

Component	Fruit Collection Site			
	Zielona Góra	Warsaw	Koszalin	Cracow
Macronutrients (g/100 g of fresh weight)				
Proteins	1.79 ± 0.02 ^a^	3.80 ± 0.20 ^c^	3.03 ± 0.10 ^b^	1.95 ± 0.17 ^a^
Lipids	1.39 ± 0.12 ^b^	0.79 ± 0.04 ^a^	3.55 ± 1.24 ^c^	3.54 ± 1.11 ^c^
Carbohydrates	18.49 ± 0.13 ^a^	19.06 ± 0.16 ^a^	19.30 ± 0.22 ^a^	18.43 ± 0.10 ^a^
Glucose	2.73 ± 0.09 ^b^	3.03 ± 0.12 ^c^	2.23 ± 0.12 ^a^	2.22 ± 0.02 ^a^
Fructose	5.78 ± 0.19 ^a,b^	6.43 ± 0.27 ^b^	5.55 ± 0.33 ^b^	5.36 ± 0.04 ^a^
Sucrose	2.28 ± 0.07 ^c^	1.65 ± 0.05 ^b^	0.92 ± 0.03 ^a^	2.65 ± 0.09 ^d^
Moisture (%) *	77.90 ± 0.09 ^c^	75.89 ± 0.0.07 ^b^	73.63 ± 0.08 ^a^	75.71 ± 0.11 ^b^
Dry matter (%) *	22.10 ± 0.06 ^a^	24.11 ± 0.17 ^b^	26.37 ± 0.05 ^c^	24.29 ± 0.03 ^b^
Ash (%) *	0.43 ± 0.01 ^b^	0.46 ± 0.01 ^b,c^	0.49 ± 0.01 ^c^	0.37 ± 0.02 ^a^
Energy value (kcal)	93.63 ± 0.15 ^a^	98.55 ± 0.21^b^	121.27 ± 0.24 ^d^	113.38 ± 0.12 ^c^

^a, b c, d^—values baring the same superscripts in rows do not differ significantly (*p* < 0.05) from each other. *—from flowering (March–April) until the end of red arils maturation (September–October).

**Table 4 molecules-26-00723-t004:** Amino acids composition (mg/100 g of fresh weight) of red arils (means ± SE, *n* = 9).

Amino Acids	Fruit Collection Site			
	Zielona Góra	Warsaw	Koszalin	Cracow
Essential amino acids				
Histidine	80.0 ± 0.0 ^c^	90.0 ± 0.0 ^d^	43.3 ± 3.3 ^a^	50.0 ± 0.0 ^b^
Threonine	80.0 ± 0.0 ^b^	110.0 ± 0.0 ^c^	80.0 ± 0.0 ^b^	73.3 ± 3.3 ^a^
Tyrosine	40.0 ± 0.0 ^b^	53.3 ± 3.3 ^c^	30.0 ± 0.0 ^a^	30.0 ± 0.0 ^a^
Valine	130.0 ± 0.0 ^c^	166.7 ± 3.3 ^d^	106.7 ± 3.3 ^b^	90.0 ± 0.0 ^a^
Methionine and cysteine	100.0 ± 0.0 ^a^	100.0 ± 0.0 ^a^	100.0 ± 0.0 ^a^	100.0 ± 0.0 ^a^
Lysine	110.0 ± 0.0 ^a^	170.0 ± 0.0 ^c^	143.3 ± 3.3 ^b^	110.0 ± 0.0 ^a^
Isoleucine	100.0 ± 0.0 ^c^	120.0 ± 0.0 ^d^	73.3 ± 3.3 ^b^	63.3 ± 3.3 ^a^
Leucine	183.3 ± 3.3 ^c^	220.0 ± 0.0 ^d^	133.3 ± 3.3 ^b^	123.3 ± 3.3 ^a^
Phenylalanine	173.3 ± 3.3 ^c^	176.7 ± 3.3 ^c^	70.0 ± 0.0 ^a^	80.00 ± 0.0 ^b^
Total	996.6 ± 0.7 ^c^	1206.7 ± 1.0 ^d^	779.9 ± 1.7 ^b^	719.9 ± 1.0 ^a^
Non-essential amino acids				
Aspartic acid	90.0 ± 0.0 ^a^	143.0 ± 3.3 ^c^	147.0 ± 3.3 ^c^	113.0 ± 3.3 ^b^
Serine	170.0 ± 0.0 ^b^	236.7 ± 3.3 ^c^	153.3 ± 3.3 ^a^	153.3 ± 3.3 ^a^
Glutamic acid	200.0 ± 0.0 ^a^	510.0 ± 5.7 ^d^	316.7 ± 6.7 ^c^	283.3 ± 3.3 ^b^
Glycine	126.7 ± 3.3 ^c^	146.7 ± 3.3 ^d^	80.0 ± 0.0 ^a^	90.0 ± 0.0 ^b^
Arginine	70.0 ± 0.0 ^a^	100.0 ± 0.0 ^c^	90.0 ± 0.0 ^b^	90.0 ± 0.0 ^b^
Alanine	110.0 ± 0.0 ^a^	226.7 ± 3.3 ^c^	236.7 ± 6.7 ^c^	203.3 ± 3.3 ^b^
Proline	150.0 ± 0.0 ^a^	220.0 ± 0.0 ^b^	210.0 ± 5.8 ^b^	153.3 ± 3.3 ^a^
Total	916.7 ± 0.5 ^a^	1583.1 ± 2.7 ^d^	1233.7 ± 3.7 ^c^	1086.2 ± 2.4 ^b^

^a, b c, d^—values baring the same superscripts in rows do not differ significantly (*p* < 0.05) from each other.

**Table 5 molecules-26-00723-t005:** Fatty acid composition (g/100 g of FA) of lipids of red arils (means ± SE, *n* = 9).

Fatty Acid	Fruit Collection Site			
	Zielona Góra	Warsaw	Koszalin	Cracow
**SFAs**	33.33 ± 0.14 ^a^	38.39 ± 0.15 ^b^	33.12 ± 0.31 ^a^	38.51 ± 0.04 ^b^
10:0	0.05 ± 0.00 ^a^	0.12 ± 0.01 ^b^	0.07 ± 0.00 ^a^	0.13 ± 0.00 ^c^
12:0	0.50 ± 0.01 ^b^	1.13 ± 0.01 ^c^	0.30 ± 0.02 ^a^	0.51 ± 0.02 ^b^
14:0	9.84 ± 0.06 ^b^	10.76 ± 0.05 ^d^	6.76 ± 0.04 ^a^	10.39 ± 0.07 ^c^
16:0	20.43 ± 0.10 ^a^	22.66 ± 0.29 ^b^	22.37 ± 0.31 ^b^	24.37 ± 0.10 ^c^
17:0	0.14 ± 0.01 ^c^	0.12 ± 0.01 ^b^	0.13 ± 0.00 ^b,c^	0.08 ± 0.00 ^a^
18:0	1.88 ± 0.02 ^a^	2.62 ± 0.07 ^c^	3.37 ± 0.03 ^d^	2.19 ± 0.03 ^b^
22:0	0.16 ± 0.00 ^a^	0.18 ± 0.00 ^d^	0.11 ± 0.01 ^c^	0.24 ± 0.01 ^b^
24:0	0.32 ± 0.02 ^b^	0.81 ± 0.03 ^d^	0.00 ± 0.00 ^a^	0.59 ± 0.01 ^c^
**MUFAs**	10.71 ± 0.07 ^c^	8.71 ± 0.08 ^b^	13.04 ± 0.04 ^d^	7.69 ± 0.06 ^a^
14:1	0.13 ± 0.00 ^a^	0.16 ± 0.02 ^c^	0.15 ± 0.00 ^b^	0.26 ± 0.01 ^d^
16:1Δ7	0.12 ± 0.01 ^a^	0.27 ± 0.00 ^b^	0.12 ± 0.00 ^a^	0.35 ± 0.01 ^c^
16:1Δ9	0.20 ± 0.01 ^b^	0.29 ± 0.00 ^d^	0.26 ± 0.00 ^c^	0.16 ± 0.00 ^a^
17:1	0.11 ± 0.01 ^a^	0.85 ± 0.03 ^c^	0.14 ± 0.01 ^a^	0.28 ± 0.02 ^b^
18:1Δ9	9.52 ± 0.06 ^b^	6.65 ± 0.08 ^a^	12.35 ± 0.06 ^c^	6.59 ± 0.10 ^a^
18:1Δ11	0.39 ± 0.02 ^c^	0.37 ± 0.03 ^c^	0.00 ± 0.00 ^a^	0.07 ± 0.00 ^b^
20:1Δ9	0.24 ± 0.01 ^d^	0.11 ± 0.00 ^c^	0.04 ± 0.00 ^b^	0.00 ± 0.00 ^a^
***n-3*-PUFAs**	19.19 ± 0.17 ^a^	25.85 ± 0.11 ^c^	24.18 ± 0.20 ^b^	28.19 ± 0.17 ^d^
18:3Δ9,12,15	18.53 ± 0.18 ^a^	25.18 ± 0.14 ^c^	23.43 ± 0.23 ^b^	26.50 ± 0.19 ^d^
20:3Δ11,14,17	0.67 ± 0.03 ^a^	0.67 ± 0.01 ^a^	0.76 ± 0.02 ^b^	1.69 ± 0.02 ^c^
***n-6*-PUFAs**	33.81 ± 0.16 ^c^	24.24 ± 0.20 ^b^	25.19 ± 0.16 ^b^	21.00 ± 0.13 ^a^
18:2Δ9,12	30.92 ± 0.12 ^c^	20.99 ± 0.22 ^b^	21.33 ± 0.11 ^b^	19.40 ± 0.11 ^a^
18:3Δ6,9,12	0.63 ± 0.03 ^b^	0.71 ± 0.01 ^c^	0.80 ± 0.02 ^d^	0.50 ± 0.02 ^a^
20:2Δ11,14	0.16 ± 0.01 ^a^	0.14 ± 0.00 ^a^	0.16 ± 0.00 ^a^	0.62 ± 0.02 ^b^
22:2Δ13,16	1.62 ± 0.03 ^b^	1.96 ± 0.01 ^c^	2.33 ± 0.04 ^d^	0.07 ± 0.00 ^a^
20:4Δ8,11,14,17	0.48 ± 0.02 ^b^	0.44 ± 0.01 ^a,b^	0.57 ± 0.02 ^c^	0.41 ± 0.01 ^a^
**PMI-FAs**	1.41 ± 0.01 ^b^	1.32 ± 0.01 ^b^	1.08 ± 0.00 ^a^	0.98 ± 0.01 ^a^
18:3Δ5, 9,12	0.09 ± 0.01 ^a^	0.14 ± 0.00 ^b^	0.08 ± 0.00 ^a^	0.23 ± 0.01 ^c^
20:3Δ5,11,14	1.25 ± 0.03 ^d^	0.94 ± 0.02 ^c^	0.82 ± 0.02 ^b^	0.62 ± 0.02 ^a^
20:4Δ5,11,14,17	0.06 ± 0.00 ^b^	0.24 ± 0.00 ^b^	0.17 ± 0.00 ^a^	0.12 ± 0.00 ^c^
**PUFAs**	54.41 ± 0.23 ^c^	51.40 ± 0.12 ^b^	50.45 ± 0.16 ^a^	50.16 ± 0.08 ^a^
NI	1.55 ± 0.32 ^a^	1.50 ± 0.21 ^a^	3.38 ± 0.45 ^b^	3.63 ± 0.15 ^b^

^a, b c, d^—values bearing the same superscripts in rows do not differ significantly (*p* < 0.05) from each other. Abbreviations: PMI-FAs—polymethylene-interrupted FAs; 18:3Δ5,9,12—pinolenic acid; 20:3Δ5,11,14—sciadonic acid; 20:4Δ5,11,14,17—juniperonic acid; NI—not identified.

**Table 6 molecules-26-00723-t006:** The content of important fatty acids of red arils (mg/100 g of fresh weight; means ± SE, *n* = 9).

Fatty Acid	Fruit Collection Site			
	Zielona Góra	Warsaw	Koszalin	Cracow
*n-3*-PUFAs	267.79 ± 1.73 ^b^	204.19 ± 0.85 ^a^	858.51 ± 6.77 ^c^	997.81 ± 5.12 ^d^
18:3Δ9,12,15	257.52 ± 2.05 ^b^	198.90 ± 0.91 ^a^	831.65 ± 6.62 ^c^	937.98 ± 5.43 ^d^
20:3Δ11,14,17	9.27 ± 0.33 ^b^	5.29 ± 0.06 ^a^	26.86 ± 0.54 ^c^	59.83 ± 0.60 ^d^
*n-6*-PUFAs	469.96 ± 1.35 ^b^	191.47 ± 1.46 ^a^	894.36 ± 3.86 ^d^	743.40 ± 3.74 ^c^
18:2Δ9,12	429.74 ± 1.31 ^b^	165.82 ± 1.44 ^a^	757.33 ± 3.28 ^d^	686.64 ± 3.08 ^c^
18:3Δ6,9,12	8.76 ± 0.36 ^b^	5.61 ± 0.04 ^a^	28.40 ± 0.60 ^d^	17.70 ± 0.44 ^c^
20:2Δ11,14	2.27 ± 0.08 ^b^	1.08 ± 0.02 ^a^	5.80 ± 0.10 ^c^	21.83 ± 0.59 ^d^
22:2Δ13,16	22.56 ± 0.36 ^c^	15.51 ± 0.04 ^b^	82.72 ± 1.10 ^d^	2.60 ± 0.10 ^a^
***n-6*/*n-3***	**1:1.7 ^d^**	**1:1 ^b^**	**1:1 ^b^**	**1:0.8 ^a^**
PMI-Fas *	19.55 ± 0.46 ^b^	10.43 ± 0.10 ^a^	38.22 ± 0.59 ^d^	34.57 ± 0.42 ^c^
18:3Δ5,9,12	1.25 ± 0.07 ^a^	1.13 ± 0.02 ^a^	2.96 ± 0.10 ^b^	8.14 ± 0.33 ^c^
20:3Δ5,11,14	17.42 ± 0.36 ^b^	7.43 ± 0.13 ^a^	29.23 ± 0.42 ^d^	22.07 ± 0.42 ^c^
20:4Δ5,11,14,17	0.88 ± 0.04 ^a^	1.87 ± 0.06 ^b^	6.04 ± 0.17 ^d^	4.37 ± 0.10 ^c^
Total PUFAs	756.30 ± 3.26 ^b^	406.09 ± 0.94 ^a^	1791.09 ± 5.78 ^c^	1775.78 ± 2.84 ^c^

^a, b c, d^—values bearing the same superscripts in rows do not differ significantly (*p* < 0.05) from each other. Abbreviations: PMI-FAs—polymethylene-interrupted FAs, 18:3Δ5,9,12—pinolenic acid, 20:3Δ5,11,14—sciadonic acid and 20:4Δ5,11,14,17—juniperonic acid. *—from flowering (March–April) until the end of red arils maturation (September–October).

**Table 7 molecules-26-00723-t007:** Elements composition of red arils (mean ± SE, *n* = 9).

Element	Fruit Collection Site			
	Zielona Góra	Warsaw	Koszalin	Cracow
Macroelements (mg/100 g of fresh weight)				
K	772.29 ± 26.27 ^a^	878.38 ± 5.14 ^b^	838.18 ± 41.19 ^a^	789.62 ± 50.86 ^a^
P	95.96 ± 1.76 ^a^	109.35 ± 6.14 ^a^	101.03 ± 6.24 ^a^	97.89 ± 7.08 ^a^
S	29.13 ± 1.38 ^a,b^	30.99 ± 2.31 ^b^	27.78 ± 1.07 ^a,b^	24.97 ± 0.52 ^a^
Ca	20.75 ± 0.49 ^a^	21.05 ± 0.53 ^a^	23.69 ± 0.09 ^b^	19.88 ± 0.33 ^a^
Mg	19.53 ± 0.28 ^a^	25.71 ± 0.58 ^c^	22.88 ± 0.47 ^b^	24.63 ± 0.34 ^c^
Na	1.12 ± 0.20 ^a^	4.90 ± 1.26 ^c^	2.81 ± 0.20 ^b^	0.86 ± 0.39 ^a^
Microelements (μg/100 g of fresh weight)				
Zn	1506.54 ± 16.98 ^b^	947.72 ± 32.46 ^a^	1146.59 ± 89.23 ^a^	1080.24 ± 56.71 ^a^
Fe	1111.39 ± 28.67 ^a^	1447.57 ± 80.42 ^b^	2537.13 ± 142.95 ^c^	976.19 ± 68.97 ^a^
B	619.40 ± 2.61 ^b^	1152.07 ± 11.10 ^c^	463.87 ± 63.22 ^a^	652.87 ± 50.63 ^b^
Cu	240.53 ± 1.24 ^b^	251.23 ± 24.40 ^a^	206.10 ± 19.21 ^a^	201.84 ± 16.15 ^a^
Mn	103.87 ± 1.93 ^b^	76.45 ± 2.62 ^a^	521.09 ± 5.35 ^c^	721.55 ± 13.56 ^d^
Cr	15.42 ± 0.64 ^b,c^	16.36 ± 1.60 ^c^	13.57 ± 1.16 ^b,c^	11.18 ± 0.53 ^a^
Mo	11.06 ± 0.43 ^a^	17.68 ± 0.28 ^b^	11.43 ± 0.03 ^a^	17.85 ± 0.48 ^b^
Co	7.34 ± 0.21 ^b^	0.13 ± 0.03 ^a^	7.53 ± 0.09 ^b^	6.03 ± 0.17 ^b^
Metals (μg/100 g of fresh weight)				
Al	466.48 ± 24.80 ^b^	616.56 ± 17.23 ^c^	1855.06 ± 148.40 ^d^	306.63 ± 7.16 ^a^
Ni	129.52 ± 1.68 ^a^	98.93 ± 8.02 ^a^	409.77 ± 4.23 ^c^	217.53 ± 4.87 ^b^
Bi	62.65 ± 0.29 ^b,c^	54.15 ± 14.44 ^b^	46.02 ± 21.56 ^b^	22.17 ± 17.43 ^a^
Ba	37.21 ± 1.23 ^b^	34.18 ± 1.79 ^b^	77.64 ± 12.40 ^c^	24.63 ± 0.41 ^a^
In	33.40 ± 2.48 ^b^	23.92 ± 3.18 ^a^	19.24 ± 2.84 ^a^	22.52 ± 2.79 ^a^
Ti	32.55 ± 3.35 ^b^	47.67 ± 1.03 ^c^	71.12 ± 3.81 ^d^	17.03 ± 2.26 ^a^
Li	13.40 ± 0.53 ^c^	22.02 ± 0.46 ^d^	10.42 ± 0.58 ^b^	3.16 ± 0.29 ^a^
Ag	14.32 ± 1.50 ^a^	12.85 ± 0.93 ^a^	11.80 ± 0.65 ^a^	12.17 ± 0.54 ^a^
Cd	7.41 ± 2.46 ^b^	9.21 ± 1.09 ^b^	4.78 ± 0.52 ^a^	24.66 ± 3.48 ^c^
Ga	7.29 ± 0.34 ^b^	11.62 ± 0.90 ^c^	16.21 ± 1.14 ^d^	2.40 ± 0.61 ^a^

^a, b c, d^—values bearing the same superscripts in rows do not differ significantly (*p* < 0.05) from each other.

## Data Availability

The data presented in this study are available in Appendix A.

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
