# Peer review of "Red Arils of Taxus baccata L.—A New Source of Valuable Fatty Acids and Nutrients"

_molecules, 2021, doi:10.3390/molecules26030723_

Round 1
Reviewer 1 Report
The submitted manuscript is devoted to the determination of the content of fatty acids, amino acids, some elements, proteins and carbohydrates in red arils of Taxus baccata L., grown in diverse locations in Poland.
The main problems of the study:
1.The composition of many yew species including Taxus baccata is well studied and many articles have been published therefore, the main problem in this paper is a poorly pronounced novelty. Perhaps that is why the authors practically avoid comparing their results with those reported earlier for Gymnosperms.
2. Moreover, there is a serious problem with the identification and determination of fatty acids.
a. It is generally accepted that only fatty acids with a regular position of double bonds (methylene interrupted) are classified as n-3 and n-6 polyunsaturated fatty acids, such as EPA, DHA, linoleic, linolenic, etc. These families do not include polyunsaturated fatty acids with an irregular position of double bonds even having a double bond in the n-3 or n-6 position. They are called non-methylene interrupted or Δ5-olefinic acids in Gymnosperms.
b. The obtained results differ principally from the data reported for fatty acids of Taxaceae (including T. baccata) and other Coniferophytes. Particularly surprising is the presence of EPA, 20:3n-3 and also 20:3Δ7,11,14; these fatty acids have never been detected before in yew berries. The authors found the abundance of linolenic acid (up to 26%), whereas it is usually a minor component (about 1-2%) in Taxus species. The authors found an abundance of linolenic acid (up to 26%), whereas it is a minor component (about 1-2%) in Taxus species. At the same time, the specific for gymnosperms 18:2Δ5,9 was not identified. These singularities appear to be the result of errors in the identification of fatty acids.
3.The abundance of stylistic errors leads to a distortion of meaning. Some of them I give in the review and in the manuscript. The authors should check very carefully the style of the manuscript.
Introduction
Struck out phrases (see MS) actually have nothing to do with the topic of the manuscript. I recommend to keep close to the topic.
“Specific PUFAs” What do you mean specifically?
“linoleic C18:2c9c12c (n-6)” and the names of other fatty acids . It’s too redundant. I recommend use or linoleic or C18:2c9c12c or 18:2n-6, but not together. And further throughout the text.
“However, to the best of our knowledge, data about the macro- and micro-nutrient composition of RAs are scarce” This statement needs to be supported by references. Moreover, I would recommend to cite the examples and data on the composition of arils.
“Essential PUFAs are represented by two families and their main compounds: linoleic C18:2c9c12c (n-6) and α-linolenic C18:3c9c12c15 (n-3) acids. These compounds are the precursors of eicosanoids…” It is necessary to rewrite, since in this version it turns out that it is linoleic and linolenic acids that are the precursors of eicosanoids. This is wrong.
“Linoleic acid competes with α-linolenic acid in the enzymatic conversion to their respective long-chain derivatives and incorporation into plasma membranes” Again, a stylistic flaw leads to an error, because neither linoleic nor its derivatives are incorporated into membranes, because membranes are composed of lipids.
“functional FAs” What does it mean? It is necessary to correct.
Results and discussion
In the tables and figures, it is necessary to specify exactly: is the content of substances determined for the dry or wet weight of the berries? In addition, how many replicate determinations (n = x) was performed?
There are many papers about distribution of toxic compounds, such as alkaloids and diterpenoids. It is well-known that all parts of the tree except the aril contain alkaloids (Vidakovie et al., 1991), and Vesela et al. (1999) show their variation during seasons. And these are not the only publications on this topic. These publications date back to the nineties. It confirms the lack of novelty of the submitted manuscript.
Materials and Methods
“As the free acidity level of lipids extracted from RAs did not exceed 0.4% of acidity value…” What does it mean?
“fat samples” What is it?
You should describe the conditions of FAME preparation.
References
I recommend to reduce the number of references.
Therefore in its current form, this manuscript is not suitable for publication in Molecules. The manuscript may be resubmitted after serious revision.
Author Response
Dear Reviewer,
I am sending responses as an attached document.
Please accept my best regards
Jaroslawa Rutkowska and Co-Authors

Reviewer 2 Report
I think that the paper under review deserves publication in Molecules. The topic is of interest for the journal readers and the described findings are scientifically sound. Overall, the paper is well written and I suggest correcting only few errors/inaccuracies, in order to improve the quality of the work. Despite the latter judgment, I would like suggesting the revision of some disputable statements.
The authors described throughout the paper the potential value of red arils in human nutrition. Although the result of the presented research indicate that RAs could be regarded as a new food (or food component), I think that some of the compounds/elements identified could rise some concerns if used in human nutrition. More specifically, deacitylbaccatin/baccatin/taxol, that are cytotoxic compounds present in RAs, are not welcome in foods. We can also say the same concerning the presence of cadmium salts.
Therefore, I suggest modifying the sentences dealing with the identification of taxus compounds in RAs, which are ‘too positive’. In addition, red arils of Taxus baccata have not yet acquired the GRAS status (to the best of my knowledge), which is very important for their use in foods. Few sentences on these topics should be added to the paper.
Last suggestion: please indicate how the fat samples were extracted from the RAs, before transmethylation with MeOH/H2SO4.
Author Response
Dear Reviewer,
I am sending our response as and attached document.
Please accept my best regards,
Jaroslawa Rutkowska and Co-Authors

Reviewer 3 Report
This manuscript described quantity of Taxus compounds, proximate composition, amino acids composition, fatty acids composition, and elements composition in red arils collected from four regions in Poland. All analytical data are fascinating and satisfied as an analysis report.
However, for example, there is no answer why 10-deacetylbaccatin III and baccatin III in red arils collected from Zielona Gora are the largest and why n-3- and n-6-PUFAs in red arils collected from Koszalin and Cracow are the largest?, etc. Readers cannot know the relationship between each content and each growth site characteristic in this manuscript.
Authors should make a research plan for such answers or submit it to an analysis data book.
Author Response
Dear Reviewer,
I am sending our response as an attached document.
Please accept our best regards,
Jaroslawa Rutkowska and Co-Authors

Round 2
Reviewer 1 Report
There is a clear problem with the naming of fatty acids in the manuscript.
1. The name "Δ-5-unsaturated polymethylene-interrupted FAs" is too much. "Polymethylene-interrupted or PMI" is clear.
2. The designation of the configuration of double bonds is also unnecessary, especially since you have not defined it.
3. The correct symbols of the FAs are 20:3Δ5,11,14 or 18:3Δ5,9,12. It is necessary to separate double bonds with commas, and "C" as well as space are unnecessary.
4. Do not duplicate the trivial name of the fatty acid and its symbols. For example, "pinolenic C18:3 Δ5c9c12" or "α-linolenic acid (C18:3 c9c12c15)" is too much, so you should use either α-linolenic acid or 18:3Δ9,12,15 (or 18:2n-6). Throughout the text.
5. L. Also the variant "linoleic (n-6) and α-linolenic (n-3) acids" is seems strange.
6. L. 267. What do you mean "n-3 category"?
So, I may repeat that There are abundance of mistakes and stylistic errors in the manuscript. The authors should check very carefully the the manuscript.
L. 32: I would clarify "known as unique for seeds of gymnosperms".
L. 39. Correct "α-linolenic acid"
L. 81-87. It's too much details.
Author Response
Dear Reviewer,
We are sending responses in attached document.
Please accept my best regards,
Jaroslawa Rutkowska

Reviewer 3 Report
This second version of the manuscript is a great improvement and is not be an analysis report.
Author Response
Dear Reviewer,
We are grateful for acceptation revision of our manuscript. We greatly appreciate your kindness, effort and time pressure.
Please accept my best regards,
Jaroslawa Rutkowska